Participation in wiki communities: reconsidering their statistical characterization

Tenorio-Fornés Ámbar atenorio@ucm.es 1
Arroyo Javier javier.arroyo@fdi.ucm.es 2
Hassan Samer 2 3
1 Decentralized Science, Decentralized Academy Ltd. , Madrid , Spain
2 Instituto de Tecnología del Conocimiento, Universidad Complutense de Madrid , Madrid , Spain
3 Berkman Klein Center for Internet and Society, Harvard University , Harvard , MA , United States of America
Zubiaga Arkaitz
Electronic publication date: 2022 Jan 3
Publication date: 2022
Volume: 8
Electronic Location ID: e792
Received 2021 Jul 21; Accepted 2021 Nov 1
Copyright: ©2022 Tenorio-Fornés et al.
Copyright year: 2022
Copyright holder: Tenorio-Fornés et al.
License: This is an open access article distributed under the terms of the Creative Commons Attribution License, which permits unrestricted use, distribution, reproduction and adaptation in any medium and for any purpose provided that it is properly attributed. For attribution, the original author(s), title, publication source (PeerJ Computer Science) and either DOI or URL of the article must be cited.
License URL: https://creativecommons.org/licenses/by/4.0/

Keywords: Commons-based peer production, Open collaboration, Participation inequality, Power-law, Truncated power-law, Online communities, Wiki communities

Funding: The European Research Council ERC-2017-STG grant no.: 759207 The Spanish Ministry of Science and Innovation RTI2018-096820-A-I00 This work was supported by the project P2P Models (https://p2pmodels.eu) funded by the European Research Council (ERC-2017-STG grant no.: 759207), and by the project Chain Community (grant no.: RTI2018-096820-A-I00) funded by the Spanish Ministry of Science and Innovation. The funders had no role in study design, data collection and analysis, decision to publish, or preparation of the manuscript.

==============================
Peer production online communities are groups of people that collaboratively engage in the building of common resources such as wikis and open source projects. In such communities, participation is highly unequal: few people concentrate the majority of the workload, while the rest provide irregular and sporadic contributions. The distribution of participation is typically characterized as a power law distribution. However, recent statistical studies on empirical data have challenged the power law dominance in other domains. This work critically examines the assumption that the distribution of participation in wikis follows such distribution. We use statistical tools to analyse over 6,000 wikis from Wikia/Fandom, the largest wiki repository. We study the empirical distribution of each wiki comparing it with different well-known skewed distributions. The results show that the power law performs poorly, surpassed by three others with a more moderated heavy-tail behavior. In particular, the truncated power law is superior to all competing distributions, or superior to some and as good as the rest, in 99.3% of the cases. These findings have implications that can inform a better modeling of participation in peer production, and help to produce more accurate predictions of the tail behavior, which represents the activity and frequency of the core contributors. Thus, we propose to consider the truncated power law as the distribution to characterize participation distribution in wiki communities. Furthermore, the truncated power law parameters provide a meaningful interpretation to characterize the community in terms of the frequency of participation of occasional contributors and how unequal are the group of core contributors. Finally, we found a relationship between the parameters and the productivity of the community and its size. These results open research venues for the characterization of communities in wikis and in online peer production.

Introduction

Since the emergence of online communities, one of the major topics of interest is to understand the different levels in which members participate: that is, the distribution of participation, also named distribution of work, or effort. Far from classical organizational structures, and more similar to volunteer-driven social movements, communities show an inherent participation inequality across its participants. Specifically in peer production communities, such as those in wikis and free/open source software, this issue has derived multiple research questions: the concentration of participation in an elite (Shaw & Hill, 2014; Matei & Britt, 2017; Kittur et al., 2007; Priedhorsky et al., 2007), the degree of participation inequality (Fuster Morell, 2010; Ortega, Gonzalez-Barahona & Robles, 2008; Neis & Zielstra, 2014), the characterization of who participates more (Hill & Shaw, 2013; Reagle, 2013), the process of changing user roles (Arazy et al., 2015; Preece & Shneiderman, 2009), or the evolution of participation depending on multiple factors (Vasilescu et al., 2014; Serrano, Arroyo & Hassan, 2018).

An important bulk of peer production research tends to say that the distribution of participation follows a power law. Intuitively, this means a very small number of contributors concentrates most of the participation (or work), highlighting participation inequality. Formally, a power law is a simple relationship between two variables such that one is proportional to a fixed power of the other.

In the issue at hand, i.e., participation, the two quantified dimensions are the number of contributions, and the share of people in the community that has made such number of contributions. The relationship among them is negative, that is, the higher the number of contributions, the smaller the share of contributors that has made such number of contributions. According to this idea, a small amount of contributions would be common, while larger amounts would be more rare. This fits with the assumption of participation inequality in which most members of the community tend to participate very little (occasional contributors), while a few of them account for an enormous amount of contributions (core contributors). In fact, the statement is not ungrounded, since several statistical studies focused on Wikipedia claim that the number of edits per user follow a power law distribution (Kittur et al., 2007; Stuckman & Purtilo, 2011), and other studies find similar behavior in free/open source communities (Healy & Schussman, 2003; Sowe, Stamelos & Angelis, 2008; Schweik & English, 2012; Cosentino, Izquierdo & Cabot, 2017) or other peer production communities (Wu, Wilkinson & Huberman, 2009; Wilkinson, 2008)1.

Figure 1 shows an example of the power law2. If we consider it represents a distribution for participation, the distribution models how frequent is to find a person that contributes X times. It can be seen that the frequency quickly declines as X grows, because most users only contribute a few times. However, it shows how we can find a small amount of contributors with a very high number of contributions.

Figure 1 Power law distribution.

For participation, the X axis represents the number of contributions made by a person and the Y axis the number of persons that made X contributions.

The power law implies an underlying regularity in the behavior of the phenomenon under study. In particular, the power relationship should hold independently of which particular scale we are looking at. This may not be the case in real data, where the tails may exhibit a more conservative behavior, and other distributions may suit better (Mitzenmacher, 2004).

While the power law has been considered a suitable distribution in many fields including online communities (Johnson, Faraj & Kudaravalli, 2014) and organizations (Andriani & McKelvey, 2009), recent studies in statistics challenge its apparent pervasiveness (Clauset, Shalizi & Newman, 2009; Broido & Clauset, 2019). According to these studies, power law distributions are complicated to detect because fluctuations occur in the tail of the distribution, and because of the difficulty of identifying the range over which power law behavior holds.

For some cases this difference between a power law distribution and other heavy tailed distributions may not be relevant, since the former may be enough to roughly represent the participation. However, using the power law as statistical characterization of wiki participation can lead to unrealistic predictions regarding the likelihood and the productivity of extremely active core contributors. A power law is a relationship in which a relative change in one quantity gives rise to a proportional relative change in the other quantity, independent of the initial size of those quantities. In the peer production field, the regularity of the power law would imply that the relationship that holds for the occasional contributors would be the same to that for the core members, which may be a strong assumption for a community when it comes to predicting the activity level and the frequency of core contributors. In other words, the tail of the distribution, which represents the activity of core contributors, may not have an extreme behavior as the power law suggests, i.e., the number of extremely active contributors and their productivity may not be as high. If that is the case, more conservative distributions, such as the the truncated power law, would provide a better fit. In fact, such distribution was found suitable in a comparative analysis of the ten largest Wikipedias (Ortega, 2009).

According to these premises, it seems reasonable to question the characterization of the participation in peer production as a power law, and consider other heavy-tailed distributions. Thus, we will apply the statistical tools proposed by Broido & Clauset (2019) to study peer production distributions, and more precisely participation distributions from wiki communities. The statistical tools proposed in that work provide a test to determine whether a distribution provides a better fit than another with respect to the empirical data provided. Thus, we will use them to analyze whether one candidate distribution consistently provides a better fit than the others. The candidates will be five well-known distributions, namely, the power law, three heavy-tailed distributions with a tail more conservative than the power law (truncated power law, stretched exponential and log-normal) and a non-heavy tailed distribution (exponential), following the example by Broido & Clauset (2019).

In our work, we focus on Fandom/Wikia, the largest wiki repository which provides a large and diverse sample of peer production communities. Fandom/Wikia accounts for over 300,000 wikis. However, because of constraints of the statistical methods used, which require a certain minimum of observations, we will use for our analysis the ∼6,000 wikis which have at least 100 registered contributors.

The rest of the article proceeds as follows. “Methodology and Data Collection” details the process followed to perform the statistical analysis and for the data collection. “Results of the statistical tests” shares the results of the statistical study of user contributions, and discusses its results through the explanation of series of graphs. The next section offers an analysis of the winning distribution, i.e., the truncated power law, and proposes an interpretation of its parameters and how they characterize the different wikis under study. The paper closes with some concluding remarks and future work.

Methodology and data collection

Methodology

Following Clauset, Shalizi & Newman (2009) and Broido & Clauset (2019), our study is divided in two analyses. First, in order to assess if the power law distribution is a plausible model for the given empirical data, we use the authors’ goodness of fit test. Then, we perform an exhaustive analysis in order to identify which distribution better describes each wiki within the data set. These two methods are explained in this section.

Goodness of fit

Clauset, Shalizi & Newman (2009) proposed a statistical test in order to assess if a distribution plausibly follows a power law. First, the power law distribution is used to model the data, finding its slope, or α parameter, and the minimum value from which the power law behavior is observed, or xmin parameter.

Afterwards, in order to compare the empirical data to different distributions, we create a set of comparable synthetic data sets that follow the distribution (i.e., have the same parameters). This allows us to compare the real data with the synthetic data, and see how they deviate from each other. This method is considered more accurate than comparing the deviation with an ideal distribution which real data may never fit. Thus, we artificially create 100 synthetic data sets per wiki, for each of the five distributions.

Thus, the distance of the real data to its power law model is compared with the distance of the synthetic data sets to their power law models. Note that the synthetic data sets are also fit to power law models to compete in similar conditions These distances are calculated using the Kolmogorov–Smirnov (KS) statistic. The goodness-of-fit test returns a p-value between 0 and 1 representing the number of synthetic data set fits that outperformed the real data fit. E.g., a p-value of 0.4 represents that the real data fits the power law better than 40% of the synthetically generated data. This p-value is then used to decide whether to rule out the hypothesis of the data following a power law. In our study, we rule out the power law model hypothesis if the p-value is smaller than 0.1, as Clauset, Shalizi & Newman (2009) and Broido & Clauset (2019) do, i.e., if the probability of obtaining a worse fit by chance is smaller than 10%. The number of synthetic data sets used to calculate the p-value determines the accuracy of the result. Following Clauset, Shalizi & Newman (2009), for the result to be accurate to within ɛ, we should generate about ɛ−2/4 samples. Our study generates 100 synthetic data sets per test, therefore, the results are within an ɛ of 0.05.

When the number of observations is relatively small, this goodness of fit test cannot rule out a power law model in those cases in which the data follows other distributions such as the log-normal or exponential. For instance, for data following an exponential distribution with λ = 0.125, at least 100 observations are needed for the average p-value to drop bellow our threshold of 0.1, while for data following a log-normal distribution with μ = 0.3, the average p-value drops below 0.1 from around 300 observations (Clauset, Shalizi & Newman, 2009). Thus, high p-values in these distributions with small number of observations should not be interpreted as the data following a power law. Moreover, as studied in the following section, even if a distribution plausibly follows a power law, other distributions may fit the data better.

This work considers wikis with more than 100 observations (i.e., wikis with over 100 registered contributors) for the p-value study for two reasons. First, as already mentioned, the goodness-of-fit test would not be able to rule out the power law. Second, as the wikis with less than 100 contributors represent more than 98% of wikis (See “Methodology and Data Collection”), the percentage of wikis passing the test due to the small number of observations may further obfuscate the result about the adequacy of the power law.

Summarizing, our study considers distributions with more than 100 observations (i.e., wikis with over 100 registered contributors), performs the goodness-of-fit tests proposed by Clauset, Shalizi & Newman (2009) considering those with a p-value greater or equal to 0.1(±0.0158)3 to plausibly follow a power law. See “Results of the statistical tests” for more details.

This study was performed using the poweRlaw R package (Gillespie, 2014). Besides, the R script source code, required for applying these statistical tests to our data, is available as free/open source software to facilitate replication4.

Likelihood-ratio test

The previously described goodness of fit test provides a tool to decide whether to rule out a power law distribution as a good model for the data. However, even if a power law model is not rejected, there may be better alternative distributions. The likelihood-ratio test allows us to compare the likelihood of the empirical data fitting two competing distributions. That is, it establishes which distribution is more likely to fit the data, and whether the difference is significant.

Following the approach described by Clauset, Shalizi & Newman (2009), our study compares the likelihood of 5 different skewed distributions. Our hypothesis is that the power law is too “ambitious” for the observations of the tail. We also expect the distribution to be heavy tailed, i.e., with a decrease of the tail slower than in an exponential distribution. In addition to these two distributions that frame the expected tail of our data, our study adds three skewed distributions that would lie in between, presenting a slower decrease in the tail than the exponential but a stronger decrease than the power law: the truncated power law (also named power law with exponential cut-off), the log-normal and the stretched exponential. Both the truncated power law and the log-normal distributions have two terms, while the power law, exponential and stretched exponential have only one. The number of terms of the distributions is relevant, since it is a factor for fitness.

The study exhaustively compares, for each wiki, the fit of the data to those five skewed distributions (power law, truncated power law, log-normal, exponential and stretched exponential), and identifies when the likelihood differences are statistically significant. It uses the Vuong method (Vuong, 1989), which considers the variance of the data, and returns a p-value that states if the likelihood differences may be due to the data fluctuations, or are significant in order to favor one distribution over the other5. As Clauset, Shalizi & Newman (2009), we consider significant the differences with a p-value smaller than 0.1, i.e., those that have less than 10% probabilities of being a result of the data fluctuations. Additionally, in order to avoid over-fitting to the tail of the distribution, we force the method to fit every contributor with at least 10 contributions. If we do not impose this condition, the method could exclude many contributors in order to find a better fit for the most active contributors, for instance a fit for the people with more than 500 contributions.

This study was performed using the Powerlaw Python package (Alstott, Bullmore & Plenz, 2014). Similar to the previous subsection, the Python script source code, required for the performed analysis, is available as free/open source software to facilitate replication6.

Data collection

This work investigates the distribution of participation in wikis from Wikia/Fandom studying the number of edits per user. Wikia/Fandom is a suitable research object to draw conclusions about participation in wikis in general. As argued by Shaw & Hill (2014), Wikia is an ideal setting in which to study peer production. Wikia only hosts publicly accessible, openly-licensed, volunteer-produced, peer production projects. To date, it is the largest and most diverse repository of open knowledge peer production, with a rich ecosystem of a broad diversity of topics, languages, community and wiki sizes. Furthermore, Wikia never restricts viewership, nor participation (except that from spammers or vandals). Wikia hosts some of the largest and most successful wikis in multiple topics and languages, such as Marvel or Star Wars fandom wikis, LyricWiki on song lyrics, Proteins scientific wiki, or AmericanFootballDatabase.

To collect our data we used the publicly available Wikia census described by Jiménez-Díaz, Serrano & Arroyo (2018) and retrieved on the 20th of February 20187. However, as explained in the methodological section, we limit our analysis to wikis with at least 100 registered contributors which have done at least one edit, and excluding bot users.

Thus, starting from this census data, and complementing it with additional information as explained below, we have created a new data set to study the distribution of participation, i.e., which is the distribution of edits made by registered contributors, excluding bots. By only including registered contributors we exclude anonymous contributors, which can be identified by their IP address. However, it is problematic to unambiguously match the IP address to a single anonymous contributor and vice versa. Furthermore, it is also difficult to consider an anonymous contributor as a member of the wiki community.

This data set is complete, since it includes all the Wikia/Fandom wikis with at least 100 contributors which made at least one contribution, resulting in 6,676 wikis, as explained in detail below.

The mentioned Wikia census provides information of ∼300,000 wikis. However, the census does not provide information on the number of edits of each participant in each wiki. Thus, such information needs to be retrieved to complement the data set.

Therefore, in order to retrieve the required data, we need to query the API of each of the wikis hosted in Wikia. Spefically, we need to query the Special:ListUsers API endpoint that every MediaWiki wiki has8. Such Special:ListUsers page lists the information of every registered user in a given wiki, e.g., username, number of edits, groups she belongs to, or date of last edit made. A perl script was developed in order to use that endpoint and obtain the number of edits performed by each registered user. In particular, the script queries the endpoint making a request for all users. Afterwards, it filters out the bot users, removing the users belonging to the bot and bot-global groups. As with the previous scripts, this perl script source code is available as free/open source software to facilitate replication9.

The data collection was performed on November 6, 2018 and it is publicly available (https://www.kaggle.com/atenorio/wikia-participation-data-20181106). It contains information about 295, 658 wikis, since 8, 433 wikis endpoints were technically unavailable10.

This data, i.e., the census wikis with the edits information, was curated to avoid duplicates and to filter out wikis without human participation (i.e., bot only) and without statistical data provided by Wikia/Fandom. After removing them, the collection contains information about 282, 039 wikis.

The reliability of the data collected is considered high. The edit numbers are as reliable as Wikia/Fandom publicly accessible statistics are (i.e., those from the Special:ListUsers endpoint). Furthermore, we have also done a consistent effort in bot identification in order to filter them out, as they may alter the participation distribution.

For statistical reasons already explained in the methodological section, this work considers only wikis with at least 100 registered (non-bot) contributors. Thus, the number of considered wikis was further reduced to 6, 676. Hence, this is not a sample, but the observed full population of Wikia/Fandom wikis with at least 100 registered users with contributions.

Results of the statistical tests

According to the goodness of fit test described in the methodological section, the power law is a plausible distribution (i.e., it cannot be ruled out) for the 83% of the 6,676 wikis from Wikia/Fandom with at least 100 registered non-bot contributors. However, as explained in the same section, that does not mean that the power law is the best choice, since other distributions may fit the empirical data better.

Thus, we perform the likelihood-ratio test to compare the pairs of the five candidate distributions as explained above. The distributions are: power law, truncated power law, exponential, stretched exponential and log-normal. For each wiki, we perform likelihood-ratio tests comparing all the competing distributions against each other. That is, we perform 10 likelihood-ratio tests for each wiki, since there are 10 possible couples.

Figure 2 Results of the likelihood-ratio test between the five considered distributions for registered contributors.

The distributions considered are: power law (PL), truncated power law (TPL), log-normal (LN), exponential (EXP) and stretched exponential (SEXP). Each arrow from A to B has the percentage of cases in which A was superior than B. The figure shows in a darker color the arrow with the higher percentage for each pair of distributions.

Figure 2 summarizes the results of these comparisons. The figure’s pentagon apexes shows each of the five considered distributions. An arrow from distribution A to distribution B represents the percentage of wikis in which distribution A was preferred over distribution B in the likelihood-ratio test, while the opposite arrow represents the percentage of wikis where distribution B was superior to distribution A. Note in some cases, the likelihood-ratio test may be inconclusive to determine which of the two distributions is better for a given wiki, and in those cases neither A nor B is superior. It is important to remark that the test being inconclusive means that both distributions fare similarly, which could mean that both are adequate or even that both are inadequate. For the sake of clarity, the figure omits the complementary percentage where the likelihood-ratio test was inconclusive, although it can be easily calculated11.

The analysis of the figure results shows that the power law is not a strong contender, as it is rarely a more likely distribution than any of its competitors, with the exception of the exponential distribution, which is also overwhelmingly defeated by the rest of the candidates.

The defeat of the exponential distribution by all candidates means that a large tail of core contributors is clearly present in the wiki participation distributions, and thus that an exponential distribution, which is not able to represent heavy tails, is not a good candidate.

However, the power law being defeated by the rest of the heavy-tailed distributions means that the tail is not as heavy or large as a power law would predict. Hence, more moderated heavy-tailed distributions are required. This conclusion is similar to the one drawn in recent works that disprove the supposed prevalence of the power law in other domains (Clauset, Shalizi & Newman, 2009; Broido & Clauset, 2019).

Thus, a correct characterization of the distributions, in nearly all cases, lies in between the exponential and the power law distributions. Among the rest of the candidates, the truncated power law stands out, since as seen in Fig. 2, it is rarely beaten by its competitors: 2.16% against the stretched exponential, 2.08% against the log-normal, 0.18% against the exponential, and 0.04% against the power law distribution. Hence, the likelihood-ratio test clearly supports the truncated power law as the most appropriate distribution to characterize participation.

Table 1 Aggregated results of the likelihood-ratio tests for each wiki counting the cases where a candidate distribution wins all tests and loses at least one test.

Distribution	Wins all tests	Loses at least one test	
Power law	0 (0%)	2816 (42,18%)	
Truncated power law	596 (8.93%)	177 (2,65%)	
Log-normal	41 (0.61%)	1159 (17.36%)	
Stretched exponential	2 (0.03%)	1492 (22,35%)	
Exponential	0 (0%)	6578 (98.53%)	

The appropriateness of the truncated power law is better appreciated when we aggregate the results of the likelihood-ratio tests for each wiki as shown in Table 1. We count the cases where a candidate distribution won all the likelihood-ratio tests for each wiki, which means that that distribution is the right choice for that wiki. In addition, we also counted the times where a candidate distribution lost at least one test, which means that for that wiki the candidate distribution was not the best choice.

It is important to remark that only in 10 wikis (0.15%) no candidate distribution won any likelihood-ratio test which means that they all were equally good (or, more precisely, bad) candidates. We have inspected these cases and they all exhibit uncommon participation distributions.

According to Table 1, the truncated power law is significantly better than all the candidates in 596 wikis out of the 6,676, i.e., approx. 9% of the wikis considered. While the rest of the distributions fare much worse: only the log-normal and stretched exponential distributions are the best candidates in 41 and 2 wikis, respectively. The power law and the exponential are not the best candidates for any wiki, which reinforces the idea of the suitability of a heavy-tailed distribution but not as heavy as that from the power law.

According to the aggregated results in Table 1, the truncated power law is not the best or among the best candidates for only 177 wikis out of 6, 676 wikis (2.65%); more precisely in 67 wikis (1%) looses one test, in 101 wikis (1.51%) loses two tests and in 9 wikis (0.1%) loses three tests. The rest of the distributions fare much worse, e.g., log-normal can be ruled out as the best candidate in the 17.36% of the wikis and the stretched exponential in the 22.73%. This result reinforces the idea of the truncated power law being the distribution of choice when trying to characterize the participation distribution in wikis, because it seems difficult to find a better one for most of the cases.

Figure 3 Complementary cumulative distribution function of participation of a wiki and the fitted distributions.

The X axis represents the logarithm of number of edits and the Y axis the inverse cumulative relative frequency the percentage of contributors that made at least X edits in the wiki.

We show an example of participation distribution where the truncated power law won all the tests in Fig. 3. The figure shows a log–log plot of the complementary distribution function where the X axis represents the logarithm of the number of edits in the wiki and the Y axis the inverse cumulative relative frequency, i.e., the percentage of contributors that made at least X edits in the wiki. The figure displays the observations (grey squares) and the fitted distributions, i.e., the truncated power law and all the candidate distributions. The observations in the left side of the graph represent the contributors with fewer edits, while those most towards the right are the core contributors that made most edits, i.e., the tail of the participation distribution.

In this figure, first we can observe the different tails of the considered distribution. While the exponential has the most conservative tail, the power law is the one that has a heavier tail, while the rest of the distributions have a tail in between them. Regarding the data fitting, the exponential with his bounded tail is not able to model the community behavior at all. The rest of them fit the initial slope, but only the truncated power law is able to successfully grasp the tail behavior, because the others predict a heavier tail.

Note the participation distribution in Fig. 3 is one of the 9% examples in which the truncated power law wins all test. Still, as mentioned, in most of the cases (97, 35%), the Truncated power law is not defeated by any other distribution. Such cases typically correspond with participation distributions with tails that can be conveniently fitted by the truncated power law, but also by the log-normal and/or the stretched exponential. So, according to this statistical evidence, the truncated power law is in fact the most adequate distribution for wiki participation.

The statistical analysis carried out shows that the truncated power law is the best distribution to characterize the participation in wikis among those considered, as it is barely rejected and is the only proper fit in 9% of the cases. In the next section, we will interpret the parameters of this distribution in the context of participation and will relate them with the characteristical features of the wiki communities.

Analysis of the truncated power law for characterizing participation distributions

In this section, we will explore the diversity of participation distributions that are modelled by the truncated power law, but before that, we need to understand better the effect and interpretation of the parameters that define the the truncated power law.

Interpretation of the truncated power law parameters

The truncated power law is defined as a power law multiplied by an exponential: x−αe−λx. In the log–log plot, the parameter α is related to the slope of the power law function, while the parameter λ is related to the starting point and/or the steepness of the decay in the tail.

Figure 4 Complementary cumulative distribution functions in logarithmic scales of truncated power laws.

Each sub-figure plots three wikis with similar α parameter, adopting smaller values in the left plot, average values in the middle and higher values in the right. The X axis represents the logarithm of number of edits and the Y axis the inverse cumulative relative frequency the percentage of contributors that made at least X edits in the wiki.

As a result, lower alphas can be associated with a higher frequency of participation of occasional contributors. While the number of contributions increase, their frequency decreases less conspicuously than in the case of higher alphas. In other words, in communities with lower alphas the frequency of contributors with more contributions decreases less significantly.

On the other hand, higher lambdas can be associated with more pronounced deviations from the power law in the tail, which means that more active contributors are less frequent as what the power law would predict. Thus, higher lambdas relate to less inequality among active contributors than predicted by the power law.

In Fig. 4, we show the truncated power law of nine wikis with different α and λ parameters that illustrate how diverse may be the participation distributions in wikis. From left to right we show three plots each of them with three participation distributions with roughly similar α values (the alpha values grow from the left to the right plot). In each plot, we show participation distributions with similar α but with different λ values. This figure illustrates the idea that the initial slope of the distributions depends on α values, as it is steeper from the left to the right plots. Besides, in each figure we can appreciate that higher values in the λ parameter are associated with a more pronounced and earlier decay sooner, or, conversely, smaller values allow the power law relationship to prevail longer.

Figure 5 Scatter plot of the TPL-distributed wikis where the color represents the number of edits.

Figure 6 Scatter plot of the TPL-distributed wikis where the color represents the number of contributors.

Relationships of the parameters with features from the wiki communities

In this section we explore whether the α and λ parameters are related to some features from wiki communities, namely, the number of edits and the number of participants. We will use scatter plots in which each dot represents a wiki in a 2-dimensional plot. The plot axes represent the values of the α and λ parameters, and the dot is colored according to a color gradient related with the specific wiki feature. More precisely, in Fig. 5 the color represents the number of edits, and in Fig. 6, it represents the number of contributors of the wiki. For the sake of clarity, the plot will only display the wikis where the truncated power law distribution won all the likelihood-ratio tests.

The scatter plots show a cloud of dots with no clear relationship among the parameters. The relationship could be inverse, since the cloud rarely includes wikis with large α and λ values or wikis with small α and λ values. However, the variability is very high to see a clear pattern.

When studying the relationship of the parameters with the size of the community in Fig. 5, we can observe how the λ parameter seems to be inversely related to the number of edits of the wiki, as the largest wikis are distributed in the lower part of the figure and vice versa. In other words, larger wikis (those with millions of edits) have smaller lambdas, which means that the decay in the tail of their participation distributions is not as significant. It reveals that, given an alpha value, there are more core contributors than in wikis whose participation distributions have higher lambda values, and that results in more productive communities in terms of edits. On the contrary, wikis with higher lambdas have a less populated elite of core contributors which results in smaller wikis in terms of edits.

At Fig. 6, we can observe that the number of contributors of the wiki is related to the combination of both parameters, as we can see that the color gradient shifts from the upper-left towards the bottom-right corner. Participation distributions characterized by high alpha values and low lambda values belong mostly to larger wiki communities (blue dots). Those parameter values determine an extremely sharp decrease in the (relative) frequency of editors as the number of edits increases, and also a more pronounced decay on the frequency of the most active contributors. In other words, extremely unequal participation distributions can be found mostly in large wiki communities. Conversely, we can find that less unequal distributions of participation –those with low alpha and high lambda values–characterize mostly the distribution of participation of wikis with smaller communities (yellow dots).

We cannot conclude if higher inequality is cause or consequence of larger communities and vice versa. Such confirmation would require further research. However, it seems that there is a clear link between community size and participation distribution.

Furthermore, it is important to bear in mind that we are observing the participation distribution during the whole life of the wiki, that is, the aggregated effect of different communities that interacted in the wiki across time, since new contributors come and other leave, or contribute in different degrees, throughout their evolution. In fact, larger communities are usually older communities. In this sense, it would be interesting to observe how the yearly participation distribution in these wikis evolved, because the highlighted inequality could potentially be the result of the aggregation throughout the years of more egalitarian distributions of participation.

Concluding Remarks

In this work, we have critically studied the distribution of participation in wikis. We aimed to analyze Wikia/Fandom, which hosts ∼300,000 wikis. From those, we selected the 6,676 wikis with at least 100 registered contributors to perform our statistical analysis. This is considered an extensive and diverse population, appropriate for an analysis following the approach defined by Clauset, Shalizi & Newman (2009). According to our results, the power law is not an appropriate distribution for wiki participation, as it predicts that core contributors are more frequent and more active than the observed in these communities. This contradicts the bulk of the peer production literature, which refers to the power law as the reference distribution when discussing about contributor participation.

In our statistical analysis we have considered potential alternatives, and from these distributions, the truncated power law gives clearly the best fit with the empirical data. Consequently, it should be considered as the distribution of participation of choice when characterizing wiki communities. Of course, it may not be adequate for some specific communities, and yet it has been able to characterize effectively the vast majority of them, while the other candidates performed significantly worse. These findings have implications that can inform a better modeling of participation in peer production, and help to produce more accurate predictions of the tail behavior, that is, predictions about the frequency and the activity level of the core contributors.

In our analysis, we have also found that the parameters of the truncated power law distribution (that govern the slope and the decay of the power law relationship in a wiki project) are related with the number of members in the community and the number of edits in the project. However, the reasons behind these findings deserve deeper consideration and are a matter of future research.

The prevalence of the truncated power law as the distribution of choice for characterizing the participation distribution in wikis has several implications. For instance, it means that the truncated power law fits better, especially concerning the frequency and the activity level of the core contributors. The change of slope of the truncated power law may also serve to empirically determine a clear division between core and non-core contributors instead of using arbitrary divisions as in other studies (Kittur et al., 2007). Further research may provide insights on how and why the inner dynamics change, and how we can study better the different emergent roles within peer production communities.

In a truncated power law, the frequency and activity level of core contributors, i.e., the highly active members, is smaller than that predicted by a power law with the same slope. That means that, when looking at the distribution tail, we can observe a sharper decrease in the frequency of extremely active contributors as the edit activity increases.

The reasons behind this fact need to be determined. They could be related with community dynamics such as some kind of elitism that prevents more people to be involved as much as those more active in the community, or that many active contributors experiment a burnout at some point and cease or decrease their activity level (Jiang et al., 2018), or even with the fact that it is not possible to find people as productive as a power law distribution predicts for certain participation levels.

Still, the difference in the participation level between core and non-core contributors is remarkable and it seems to reinforce the idea that core contributors are somehow special, in the sense that there is a qualitative change in their work and motivations (Burke & Kraut, 2008) and thus higher barriers to join them, and/or the elitization of the core leads to oligarchies (Shaw & Hill, 2014).

The approach followed by this work has several limitations. It is a descriptive quantitative work, and thus it lacks explanatory aspects that further qualitative research could contribute with. Besides, we are cautious with the generalizability of our findings beyond Wikia/Fandom, i.e., to every wiki communities or to peer production communities in general. That is, could we argue that the distribution of participation in peer production is a truncated power law? We cannot prove that empirically, and yet we have a good base for cautious claims in that regard; similar to other generalizations performed in the field, e.g., by Shaw & Hill (2014). That is, considering the significant size and diversity of the sample used, there is good evidence for potential generalizability. In order to support this generalization, these results would need to be validated in other projects, the such as the Wikimedia Foundation projects, as well as in other peer production communities such as Free/Open Source Software projects. Thus, we encourage other researchers to replicate our approach with other peer production communities.

Furthermore, the statistical analysis methods employed require a certain number of observations to have conclusive results, which constrains their applicability for studying the participation distribution of wikis with small communities. Despite of having near 300,000 wikis in Wikia, most of them have under 100 registered contributors and were discarded, using “only” 6,676 wikis in the analysis. For wikis with smaller communities statistical methods may find difficult to provide conclusive results as the differences are subtle and mostly related with the tail behavior.

We have analyzed the participation in the communities aggregated through time (years), that is, accumulating the participation of all the members from the beginning. However, the members of a wiki community change through time, as change the participation dynamics. The participation distribution could be different when analyzed in a smaller time window, such as a year.

We have already defined several potential lines for future work, but we would like to mention those that we consider more interesting. First, it would be relevant to use a different base population, in order to appropriately generalize for peer production communities and not just wikis. For instance, we could analyze in a similar manner communities from Github, Wikimedia Foundation projects, or Stack Exchange. Second, it would be useful to perform a temporal analysis with a rolling time window, in order to understand how these distributions evolve over time. This is especially relevant if we consider the evolution of the truncated power law parameters and how they relate with participation dynamics and inequality. In fact, we can highlight the importance to deepen the study the characterization of wikis based on their truncated power law parameters. That is, it would be interesting to cluster similar wikis and explain the causes or consequences of the different typologies. Moreover, we could explore how they relate with factors such as maturity stage, community dynamics and sustainability.

Our work asserts the truncated power law is probably the most appropriate distribution to represent the distribution of participation in wikis from Wikia. Our results can be better understood if they are observed in the context of a previous study that questioned the prevalence of power law in several fields (Clauset, Shalizi & Newman, 2009) and the ground-breaking finding that the power law was indeed rare in real-life networks (Broido & Clauset, 2019). Our finding will thus open new lines of research, revisiting old assumptions in the field, exploring further the causes behind the observed structural change in core contributor participation and the relationships with the sizes of the community and the project and other factors behind the behavior.

Supplemental Information

Supplemental Information 1 Data set with the edits per registered user from the Wikia/Fandom wikis

The csv file uses ’;’ as delimiter instead of ’,’, but it is a text file that can be read and parsed normally. It can also be found at Kaggle: https://www.kaggle.com/atenorio/wikia-participation-data-20181106.

Click here for additional data file.

Supplemental Information 2 Code to retrieve the edits of the Wikia users

Click here for additional data file.

Supplemental Information 3 Code to perform the statistical tests

Click here for additional data file.

Additional Information and Declarations

Competing Interests

Author Contributions

Data Availability

1 Other studies just mention a highly skewed distribution or similar statements without further specification (Howison, Inoue & Crowston, 2006; Crowston et al., 2006; Barbrook-Johnson & Tenorio-Forns, 2017).

2 Original picture by Hay Kranen PD. available at Wikimedia Commons. Our version is a slight variation from the original one.

3 The confidence interval is due to the test resolution that depends on the number of synthetic data sets considered.

4 Goodness of fit tests script: https://github.com/atfornes/WikiaDistComparison/blob/master/p-value.r.

5 The method is adapted by Clauset et al.’s for nested distributions such as power law and truncated power law, where a family of distributions is a subset of the other. Such modified method, which we use as well, allows to state whether the larger family is indeed needed or both distributions are good models.

6 Likelihood-ratio test script: https://github.com/atfornes/WikiaDistComparison/blob/master/powerLawVsPowerExp.py.

7 Wikia census: https://www.kaggle.com/abeserra/wikia-census.

8 Note all Wikia/Fandom wikis use the same wiki software, MediaWiki, the maintained by Wikimedia Foundation and used by its projects, including Wikipedia.

9 Script to retrieve user contributions: https://github.com/Grasia/wiki-scripts/tree/master/users_with_edits/.

10 Wikis may be unavailable for a number of reasons, e.g., being removed from the platform, or having changed their name. Unavailable wikis represent 3,5% of the total wikis, constituting a small percentage of expected noise that should not compromise the results of the study.

11 In all cases, percentage of A > B + percentage of A < B+ percentage of inconclusive = 100%.

The authors declare there are no competing interests.

Ámbar Tenorio-Fornés conceived and designed the experiments, performed the experiments, analyzed the data, performed the computation work, prepared figures and/or tables, authored or reviewed drafts of the paper, and approved the final draft.

Javier Arroyo conceived and designed the experiments, analyzed the data, prepared figures and/or tables, authored or reviewed drafts of the paper, and approved the final draft.

Samer Hassan conceived and designed the experiments, authored or reviewed drafts of the paper, and approved the final draft.

The following information was supplied regarding data availability:

The raw data and code are available in the Supplemental Files.

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
