# Peer review of "Participation in wiki communities: reconsidering their statistical characterization"

_PeerJ Computer Science, doi:10.7717/peerj-cs.792_

## Round 0.1 · original submission · Minor Revisions

The paper should become acceptable once the minor revisions suggested by reviewer #2 are addressed. Most importantly, we would like to see a revised submission where the importance and implications of the research are more clearly stated. Reviewer #2 has also outlined a set of minor revisions that should be taken into account in revising the submission, as well as an annotated pdf where more specific suggestions are given.

Reviewer 1 ·

Basic reporting

The manuscript uses appropriate language, references relevant research, and follows a standard article structure.

Experimental design

The data collection and analysis strategy is well-motivated, clearly described, builds on prior work, extends into a new domain, and is rigorously applied.

Validity of the findings

The data was appropriately retrieved and analyzed using constructs/methods, the findings replicate other empirical findings about power laws in social systems, discussion emphasizes truncated power law as appropriate fit and the corresponding mechanisms and interpretations that generate them.

Additional comments

I would add the following references into the manuscript:

Mitzenmacher, M. (2004). A Brief History of Generative Models for Power Law and Lognormal Distributions. Internet Mathematics, 1(2), 226–251. https://doi.org/10.1080/15427951.2004.10129088

Andriani, P., & McKelvey, B. (2009). Perspective—From Gaussian to Paretian Thinking: Causes and Implications of Power Laws in Organizations. Organization Science, 20(6), 1053–1071. https://doi.org/10.1287/orsc.1090.0481

·

Basic reporting

I am happy to review this article, which examines the distribution of participation in online wiki communities.

Overall, the language of this manuscript is adequate. There are a number of places where the language feels awkward to a native English speaker, but only rarely do these weaknesses hinder comprehension. I have marked some of the most egregious examples in the attached PDF. There were a few places, most notably at the end of the introduction, where \ref's were broken. It looks like this may have been because the style of the manuscript does not include section numbers?

The paper identifies the most important relevant literature and is well-situated from a methodological perspective. I think that the authors could have done more to explain the practical importance of the research and to identify more concretely the practical implications of identifying a context as having distributions which are fit better by one function versus another. In other words, what is the scientific and theoretical benefit of applying Broido and Clauset's approach to all of these new contexts?

In general, the structure of the article was effective. The figures were well done and persuasive. I found the multiple colors of Figure 1 a bit confusing. Does the cutoff point represent the mean, for example? This should be clarified, and referenced in the legend. I found Figure 2 engaging, but I wished that the edges were colored or weighted based on the percentage, to make it easier to distinguish the "winners" visually. At least in my copy, Figure 3 appears to have some compression artifacts and should be produced as a vector image (also Figures 5 and 6). Finally, I thought that Figure 4 might be more persuasive if it also included empirical data points (although it's possible that this might make the figures too noisy).

Regarding the data, the data and code include appears to be adequate for running the statistical analyses. I was unable to find the code used to actually gather the edits per person, as described in lines 195-204. Nor was this raw data made available. I did not run any of the code to test it.

Experimental design

The overall design of this paper is appropriate for a computer science journal, well defined, and well executed. The methods are well situated in previous work and well described.

As explained above, I do think that the authors could and should do more to identify the knowledge that this approach gives us that we didn't have before, especially in the Introduction / Background section.

I have only two, fairly minor suggestions for the methods section. First, the authors claim to consider only communities with 100 contributors at times, and at other times those with 100 registered users. These are different measures, and it should be clear which cutoff is the actual cutoff. If it is registered users, does that mean that unregistered users (i.e., "IP users") are also removed? If so, this should be made clear and justified. Second, the authors should explain why 8K wiki endpoints were not available. I would guess that this is because the wikis no longer existed, but that should be made clear.

Validity of the findings

The findings are well-supported and the analysis of this paper appears both reasonable and statistically sound. Overall, the authors make a convincing case that wikis are typically well-described by a truncated power law.

As mentioned above, I would have liked to have a more substantive understanding of what the conclusions infer for our understanding of how these communities operate, as well as what else we might be able to do with this approach. When these sorts of explanations occurred, I was often unconvinced. For example, the manuscript seems to argue that the findings show that high-volume contributors differ from low-volume contributors, but this seems like it would already be very likely. Indeed, if anything a power-law distribution would suggest that they are more different from each other (in the sense that the discrepancy in the number of edits is larger).

I believe that the concluding remarks section should be restructured from bullet points to paragraphs. I found the description of generalizability at ~398-401 confusing.

Additional comments

Overall, I found this paper to be convincing in showing that a truncated power law is a reasonable distribution for characterizing wikis. The analysis is narrow but well-executed and while I provided a number of suggestions for improvements to the presentation and discussion of the results, I think that the paper shows a number of strengths.

---

## Round 0.2 · accepted · Accept

The current revision addresses all of the comments raised by the reviewers in the previous round and therefore I approve it for publication in its current form.